# Subjective Perceptions of Occupational Fatigue in Community Pharmacists

**DOI:** 10.3390/pharmacy11030084

**Published:** 2023-05-09

**Authors:** Taylor L. Watterson, Michelle A. Chui

**Affiliations:** 1Department of Family Medicine and Community Health, University of Wisconsin School of Medicine and Public Health, 750 Highland Avenue, Madison, WI 53705, USA; 2Social and Administrative Sciences Division, University of Wisconsin-Madison School of Pharmacy, 777 Highland Avenue, Madison, WI 53705, USA

**Keywords:** pharmacist, community pharmacy, fatigue, mental fatigue, workplace

## Abstract

Introduction: Community retail pharmacists are experiencing unsafe levels of stress and excessive demands within the workplace. One aspect of workload stress that has been overlooked among pharmacists is occupational fatigue. Occupational fatigue is a characteristic of excessive workload including increased work demands and reduced capacity and resources to complete the work. The goal of this study is to describe the subjective perceptions of occupational fatigue in community pharmacists by using (Aim 1) a previously developed Pharmacist Fatigue Instrument and (Aim 2) semi-structured interviews. Methods: Wisconsin community pharmacists were eligible to participate in the study and recruited via a practice-based research network. Participants were asked to complete a demographic questionnaire, a Pharmacist Fatigue Instrument, and semi-structured interview. Survey data were analyzed using descriptive statistics. Interview transcripts were analyzed using qualitative deductive content analysis. Results: Totally, 39 pharmacists participated in the study. From the Pharmacist Fatigue Instrument, 50% of the participants stated they had times where they were not able to go above and beyond standard patient care on more than half of the days they worked. A total of 30% of the participants reported that they found it necessary to take short-cuts when providing patient care on more than half of the days they worked. Pharmacist interviews were separated into overarching themes including mental fatigue, physical fatigue, active fatigue, and passive fatigue. Conclusions: The findings highlighted the pharmacists’ feelings of despair and mental fatigue, fatigue’s connectedness to interpersonal relationships, and the complex nature of pharmacy work systems. Interventions aimed at improving occupational fatigue in community pharmacies should consider key themes of fatigue that pharmacists are experiencing.

## 1. Introduction

Community retail pharmacists are experiencing unsafe levels of stress and excessive demands within the workplace [1,2,3,4,5,6,7,8]. Pharmacists are crucial resources for health information as the most accessible health care professional. In particular, in rural communities with limited access to primary care providers, pharmacists are critical for the provision of quality care. In the 2019 National Pharmacist Workforce Survey, 72% of community pharmacists reported their workload levels as high or excessively high [1]. Respondents stated that workload negatively impacted their opportunity to take breaks as well as their mental and emotional health (70% and 51%, respectively). Even more alarming was that 39% of community pharmacists reported that their workload negatively affected the quality of care they were able to provide to patients. This was further aggravated during the Coronavirus Disease (COVID-19) crisis [9,10]. One aspect of workload stress that has been overlooked among pharmacists is occupational fatigue.

Occupational fatigue is a characteristic of excessive workload—both increased work demands and reduced capacity and resources to complete the work [11,12,13,14]. Beyond workload and work demands, fatigue is a product of the employee’s physical work environment, organizational factors such as policies and procedures, and even personal factors [15]. Health care professional occupational fatigue is crucial to study as it has implications for employee, organization, and patient outcomes [11,12,13,14,16,17,18,19]. Occupational fatigue is complex and multidimensional. Occupational fatigue can encompass numerous domains and can be described as mental, physical, acute, chronic, active, or passive [14,20,21]. Mental fatigue includes lack of motivation (feeling uninterested, indifferent or listless) as well as sleepiness [22]. Physical fatigue includes physical exertion and discomfort as well as a general lack of energy [23]. Acute fatigue often occurs immediately after work has been completed whereas chronic fatigue includes more longstanding fatigue experienced over time [20]. Active fatigue occurs during periods of “overload” from prolonged work, an example is driving on a busy freeway [21]. Passive fatigue occurs during periods of “underload” and low cognitive work, such as driving on a straight-highway with limited traffic. These characteristics help to describe this complex phenomenon of fatigue.

Consistent with other health care professionals, the well-being of pharmacists has often been overlooked [2,4,5,6,7,8,24]. Community pharmacies are often isolated from the patient’s health care team, both geographically and without direct access to complete information via the electronic medical records [25]. Additionally, community pharmacists have limited ability to control their rate of work as patients typically do not make appointments to fill and pick up their prescriptions. As the most accessible healthcare professional, community pharmacists must juggle patients in the pharmacy, on the phone, or even at the drive-through window. Community pharmacists often complete these tasks with other system demands including 12- to 14-h shifts, limited ability to sit for extended periods of time, and often working without scheduled breaks (even for lunch). Finally, community pharmacies are unique from other practice settings in that most are for-profit institutions (as compared to non-profit or academic health systems). This may yield competitive markets and limited transparency between companies that prioritize improving profitability over improving patient and workforce outcomes [25]. 

The COVID-19 pandemic thrust pharmacist work and well-being to the forefront of public media. As a major access site for COVID-19 testing and vaccinations, pharmacies were required to adjust and reprioritize work. Subsequently, patients were exposed to systematic changes including longer wait times, staffing shortages, and policy adjustments (e.g., closing the pharmacy for lunch breaks). Articles in the New York Times have included catchy titles such as, “How Chaos at Chain Pharmacies is Putting Patients at Risk” and “Angry Customers, More Work and Longer Hours Strain Pharmacists” [26,27,28,29].

Recently, pharmacists on social media have taken to use the tag “#PizzaIsNotWorking” to highlight pharmacy working conditions. The hashtag, a tongue-in-cheek reference to organizational support in the form of pizza as opposed to bonuses or staffing, has garnered public attention as pharmacists “band together to raise awareness about working conditions” [30,31,32,33]. 

Research literature and pharmacists themselves via social media are alluding to the significant imbalances between work demands and capacity. Given the unique and complex nature of community pharmacist work, a tailored, adapted, and context-specific instrument may be necessary to completely capture this multidimensional phenomenon [25,34,35]. Subjective fatigue measures and scales have been adapted in other situations to tailor the instrument to cultural, language, or contextual differences [34,36,37,38,39]. The benefit and utility of a reliable and context-specific community pharmacy instrument must be considered in light of its limitations—fatigue results and findings may have limited generalizability and comparability to other professions and settings. In order to develop and test interventions to reduce pharmacist fatigue, we must first have a robust understanding of the problem as well as reliable and valid measurement tools to compare changes to pharmacist fatigue over time. 

The goal of this study is to describe community pharmacist subjective perceptions of occupational fatigue using (Aim 1) a previously developed Pharmacist Fatigue Instrument and (Aim 2) semi-structured interviews.

## 2. Materials and Methods

The study was approved by the Health Sciences Institutional Review Board (IRB) at the University of Wisconsin-Madison (ID 2021-0174-CP001 on 25 May 2021) prior to study recruitment. 

### 2.1. Participants

The study aimed to recruit 60 Wisconsin licensed community pharmacists to participate in the study. Individuals who participated in a previous pilot study (n = 20) were not eligible to participate. The original sample size of 60 was chosen as an appropriate estimate to ensure results and saturation of findings. The final sample was limited due to financial and time restrictions for this project.

### 2.2. Recruitment

Subjects were identified based on their voluntary enrollment in the PearlRx (Pharmacy Practice Enhancement and Action Research Link) practice-based research network. To join PearlRx, individuals give permission to share their contact information with researchers from Wisconsin schools of pharmacy and indicate that they are willing to be contacted about possible research study participation. PearlRx is a statewide research network based in the Sonderegger Research Center of over 200 community, clinic, and hospital pharmacists. During the recruitment process, the PearlRx Program Administrator sent four emails to all PearlRx members, alerting them of a new study opportunity for community pharmacists. Pharmacists were also encouraged to forward and share the email newsletter to other interested pharmacists. The call for participation was also placed on PearlRx’s “Current Projects” page. Interested participants were screened for eligibility and asked to meet the researcher at a time and location that was convenient. Participants were instructed to document the days and times they worked the following 14 days on a provided calendar. The calendar was used to anchor the respondent’s recall period [40]. At the end of the 14 days, they were asked to complete the Pharmacist Fatigue Instrument and demographic survey, then return the calendar and surveys via a prepaid USPS box. After 14 days, participants were sent a reminder email to complete the survey and schedule a follow-up video interview. At the completion of the interview, the participants received a $100 Amazon e-gift card. 

### 2.3. Data Collection 

Demographic information collected included age, gender, title, years working as a pharmacist, years working in the current position, number of adults and minors in the household, and amount of caffeine consumed per day. The Pharmacist Fatigue Instrument was previously developed and tested to assess occupational fatigue in Wisconsin pharmacists [41]. The 12-item instrument captures mental and physical occupational fatigue domains found tailored from literature, including the Swedish Occupational Fatigue Inventory [14,41]. The Pharmacy Fatigue Instrument is available in Appendix A.

Individuals participated in a 30-min semi-structured virtual interview. The purpose of this interview was to explore factors related to fatigue during the 14-day recall period, including workload, work system characteristics, scheduling, and personal factors.

### 2.4. Data Analysis

The Pharmacist Fatigue Instrument and demographic responses were entered into a secure electronic platform and double-verified. Descriptive statistics, response frequency, and data visualizations were created using R Version 4.0.3 and R Studio [42]. 

All interviews were audio recorded and transcribed verbatim. The transcripts were analyzed using qualitative deductive content analysis using concepts from the literature to explore pharmacist perceptions and experiences of the various dimensions of occupational fatigue (QSR NVivo). The deductive codes aligned with the domains of mental fatigue, physical fatigue, and general lack of energy, as well as active fatigue and passive fatigue from the Adaptation Model of Stress and Performance [43,44]. The finalized codebook is available in Appendix A. The results were represented visually in a conceptual framework, based off of the Systems Engineering Initiative for Patient Safety (SEIPS) framework and Conceptual Model of Occupational Fatigue in Nursing [15,41,45]. One study team member (TW) coded all the transcripts and identified themes. The themes and exemplar quotes were presented to study team members to ensure face validity.

## 3. Results

During data collection, from July 2021 to March 2022, 49 individuals expressed interest in participating in the study and completed the initial recruitment survey. Of those individuals, 39 participants participated in data collection. No Pharmacist Fatigue Instrument data were missing.

Participant pharmacist and pharmacy demographic characteristics are summarized in Table 1. 

### 3.1. Pharmacist Fatigue Instrument

The Pharmacist Fatigue Instrument asked respondents to report the frequency they experienced fatigue-related events over the previous 14 days. Response distributions are represented in Figure 1. 

In addition to feelings of decreased energy, increased tiredness, and general fatigue, 52% of pharmacists reported they had times where they felt that they could not keep with their work on more than half or all the days they worked (35.0% and 17.5%, respectively). 

A total of 50% of the participants stated they had times where they were not able to go above and beyond standard patient care on more than half or all the days they worked (27.5% and 22.5%, respectively). Overall, 37.5% of the participants stated that they had trouble thinking clearly at work on more than half of the days they worked. In addition, 37.5% had times where they felt they were not performing at their best on more than half or all the days they worked, while 32.5% had times where they forgot whether they had completed a task on more than half or all the days they worked. Finally, 30% of the pharmacist participants reported they found it necessary to take short-cuts when providing patient care on more than half or all the days they worked (12.5% and 17.5%, respectively).

### 3.2. Pharmacist Interviews

Pharmacist interviews shed light on subjective experiences of fatigue. Their experiences were separated into several overarching themes, correlating to fatigue domains found in the literature: mental fatigue, physical fatigue, active fatigue, and passive fatigue. The key fatigue characteristics are represented in Figure 2, a conceptual model of occupational fatigue in pharmacists. Results from this study only pertain to the “Outcome” of Pharmacist Occupational Fatigue.

#### 3.2.1. Mental Fatigue

Mental fatigue is often described as feelings of lack of motivation (e.g., feeling uninterested, indifferent, passive, listless, and the lack of initiative) and sleepiness [22]. Participants noticed they were fatigued based on their interpersonal relationships with other staff members and patients. For example, a participant described that over the course of the day, they noticed that they would become “crankier” and more irritable with customers:


*You can see a difference in the beginning of the day how we treat customers to the end of the day, which is sad. I don’t ever want to get cranky at customers, but, understandably, they’re losing patience with us since everything’s taking longer, but we’re also losing patience with the fact that they don’t have patience with us. So, admittedly, I’ve been getting a lot more cranky at work, especially these last few weeks.*
Pharmacist 9

Another pharmacist indicated that on days with heavier workloads, the staff members are more likely to get irritated with each other:


*Throughout the day, I can tell we all become a little more irritated with each other. It also depends how the workflow is going, you have your days where nothing is going right, and then you have days where everything’s a cake walk. But, if we’re all caught up and things are going great, we all are like joking around, laughing with each other. But if things are just rough, we’re just snippy with each other, and it’s not fun.*
Pharmacist 11

Pharmacists also described their fatigue as changes to their overall mental states. Many pharmacists described their fatigue as feeling exhausted and worn out after a long day of work or as a general desire for rest. This fatigue had potential implications for relationships once they were home with family members. One pharmacist stated:


*And that does wear me out by the end to the day, I don’t want to talk to anybody. I just want to sit down and be quiet for a while. That fatigue. Not fall asleep fatigue, but just … I need rest.*
Pharmacist 1

Other pharmacists described their mental fatigue and changed mental states as feeling overwhelmed or feeling that they were unable to keep up with their work tasks.


*I definitely feel, it’s fatigue, but it’s also just almost more of a mental … despair sounds dramatic. It’s not despair. But, ‘oh, there’s so much to do, and I’m not going to get it done,’ kind of a feeling until it starts to slow down.*
Pharmacist 4

Pharmacists also described a “hazy” or “foggy” feeling that would occur later during the workday. Often participants would describe forgetting whether they had completed tasks such as checking prescriptions or what information they told patients during prescription counseling. Participants stated that they would verify the “same prescription” multiple times because they had trouble concentrating or would get distracted from the singular task. One participant described this feeling of mental fatigue as:


*As I start to get towards that end of the day, that foggy hazy feeling. Most of the time I am doing the vaccines, so it’s okay. But, I’ll start to get that foggy, hazy feeling, and I’ll be like, ‘wait, what did I just say? What was I just working on?’*
Pharmacist 1

#### 3.2.2. Physical Fatigue 

In the literature, physical fatigue includes physical exertion and physical discomfort [14,22]. Community pharmacists described physical fatigue as discomfort arising over the course of the day, most commonly in the low back, feet, or headaches. 

Pharmacists mentioned low back pain that developed over the course of the day, potentially attributable to standing in the same location for long shifts. Some pharmacists stated this pain was exacerbated by age or previous injuries. One pharmacist described their experience with low back pain as:


*Sometimes I do get low back pain. And I don’t know if it’s just from standing for ten hours straight in the same spot for most of the day.*
Pharmacist 11

Additionally, pharmacists also described lower leg or foot pain arising over the course of the day. Some participants detailed personal efforts to mitigate this pain by wearing more expensive shoes, arch support, and insoles.


*Usually, my feet will start to hurt. I’ll be walking back from the flu room or something, and I’ll just notice my feet are sore. I even got these better shoes, and that seemed to work for a little while. But now I’m like ‘ugh.’*
Pharmacist 20

Some pharmacists reported that the headaches were associated with certain tasks or simply the result of prolonged shifts without opportunities for food or rest.


*And especially if [the patient] is going to come yell at you, or you recognize the name and you know it’s a script you have to refuse, and it’s going to probably start an issue, and you have to keep smiling through the whole thing. That does make me get headaches by the end of the day.*
Pharmacist 1

#### 3.2.3. Active Fatigue

The Adaptation Model of Stress and Performance states that active fatigue results from conditions of high cognitive overload with continuous and prolonged work [21]. Pharmacists reported feelings of cognitive overload consistent with active fatigue. For example, one participant stated:


*It’s just mentally exhausting helping person after person after person, running back and forth between trying to keep up on product checking. Just physically, mentally, it’s exhausting. And then by the end of the day, my brain just doesn’t, I don’t feel as clear. It’s hard to think.*
Pharmacist 12

Commonly, pharmacists stated they would not notice these feelings of fatigue “in the moment” but rather once they had a moment to pause or stop. Pharmacists stated this was most prevalent when patient load decreased when the pharmacy was close to closing or when they got in their cars to drive home.


*I think for me, one of the signs that I’ll usually pick up on that makes me realize that I’m more tired is usually once I have a chance to actually sit down, I’ll be like, ‘oh, my God, this actually feels really good right now.’ Especially with those long days where I’m working 7:30 a.m. to 8 p.m. I have a lunch break in there, but you’re just continuously working. And then after a while, I sit in my car for a minute, and ‘wow, it’s been a long day.’*
Pharmacist 8

#### 3.2.4. Passive Fatigue

In comparison to active fatigue, the Adaptation Model of Stress and Performance states that passive fatigue is indicative of low cognitive work and under stimulation [21]. Pharmacists described experiences of passive fatigue when work load was minimal or the pharmacy was “slow” with few prescriptions or patient encounters. 


*Because days that we’re super busy, your brain doesn’t have a chance to slow down and stop, whereas a day where you’re slow, it’s hard because your brain’s not constantly stimulated. And then I feel like when you do have the sudden person come in, you’re like, ‘oh, I have to check.’ I’m one of those ones, I function better on a day where we’re consistently busy. I like where it’s like not overwhelming, but yet it’s busy enough where I never actually check out of what I’m doing to have to check back in. I think, when you are having a slow day, and there isn’t a lot going on it’s hard to keep engaged with what’s going on.*
Pharmacist 35

Even when the pharmacy was not “slow,” pharmacists described instances of passive fatigue as working on “autopilot” while completing tasks. Pharmacists noticed these feelings when they recovered and returned to focus or forgetting the details of a prescription they recently verified. Two pharmacists described their experiences with this type of passive fatigue in the following exemplar quotes:


*The scripts start to run into each other. And that’s also when you’re supposed to be checking them against other medications and stuff, and I just notice that I start to really zone out in what I’m doing. And I really have to snap myself and be like, ‘okay, you have to focus. You got to make sure this is correct.’*
Pharmacist 9


*Autopilot, I think is the way I would describe it. Because, there’s been times where once I’ve checked the prescription and I’ll pull it back up, I’ll be like ‘I literally don’t even remember looking at those pills or seeing that prescription.’ Or I guess I’d compare it to the times where I’ve driven home and I’m like ‘how did I just get home? I don’t remember turning here.’ It just goes by, and I’m like ‘what happened?’*
Pharmacist 11

## 4. Discussion

Overall, the Pharmacist Fatigue Instrument and interviews presented similar findings regarding subjective perceptions of fatigue in pharmacists. Specifically, the findings highlighted three themes related to pharmacist fatigue: feelings of mental fatigue, connection to interpersonal relationships, and complex work systems with both active and passive fatigue. 

### 4.1. “Despair” and Pharmacist Mental Fatigue

The survey instrument and interviews shed light on pharmacists’ perceptions of the mental fatigue they experienced at work. More than 60% of participants stated that their energy decreased, and they felt more tired over the course of their shifts most of the days they worked. Instrument respondents also shared their trouble thinking clearly and taking longer to complete tasks. These results are similar to the previous study, which identified mental fatigue as a key domain of pharmacist occupational fatigue [41].

The objective instrument results align with the interview findings where participants described feeling exhausted, worn out, overwhelmed, hazy, or foggy. This is similar to findings in other healthcare professionals [12,46]. A 2017 study of nurse managers described the signs and symptoms of fatigue as “a lack of focus, distractions, decreased tolerance, feeling overwhelmed, and a desire to rest” [12]. These findings also reflect the growing pharmacist initiatives on social media and media including “#Pizzaisnotworking” [26,27,30,31,32,33]. 

Exploring pharmacist work through an occupational fatigue lens helps to describe this complex and multidimensional experience. With a tailored and context-specific instrument, researchers and organizations can further study community pharmacist occupational fatigue as well as the impact of intervention over time. 

### 4.2. Fatigue and Interpersonal Relationships

During the pharmacist interviews, participants stated they knew they were fatigued when they became more irritable with team members or cranky with patients. Within the Pharmacist Fatigue Instrument, over 30% of pharmacists stated that on more than half of the days they worked, they felt more impatient later in the day than they did at the beginning. Similarly, 50% of pharmacists stated that on more than half of the days they worked, they felt they were unable to go above and beyond when providing patient care. 

The instrument and interview findings help to conceptualize the interpersonal nature of occupational fatigue. Within the literature, fatigue is often considered a “personal” phenomenon that an “individual” experiences—the body’s biological desire for rest and recovery. Efforts to address fatigue are often directed at the individual level, such as proper sleep hygiene or promoting mental health, well-being, and resiliency in the face of stress. While this individualized definition is correct, fatigue must also be considered within the larger social system [47,48]. A 2022 study by Cho et. al. explored the relationships between teamwork and occupational fatigue in hospital nurses in the United States [49]. The study identified that nursing teamwork was negatively related to acute and chronic fatigue. Furthermore, nurse teamwork factors of team orientation, shared mental models, and team leadership moderated the relationship between workload and chronic fatigue. The results obtained by Cho et. al. suggest that efforts to improve teamwork may be beneficial strategies to reduce fatigue. 

The results also shared the connection between occupational fatigue and relationships with patients. Research studies provide demonstrable evidence to the increased risk to patients when health care workers are fatigued [17,19,50,51,52,53,54]. A 2007 study Gander et. al. examined fatigue-related risk among New Zealand junior doctors working ≥ 40-h per week (n = 1366) [52]. Overall, 42% of respondents recalled a fatigue-related clinical error in the 6-months prior to completing the study’s survey. 

This research study demonstrated that fatigue is related to pharmacist interpersonal relationships with family, team members, and patients. In a study of emergency medicine consultants, one-third of participants reported that stress at work caused them to be irritable with patients, colleagues, or reduce their standard of care at least monthly [55]. Addressing the problem of fatigue requires looking beyond the “individual” to other social and technical elements of the work system. 

### 4.3. Active and Passive Fatigue—A Fine Balance in Complex Work Systems

Occupational fatigue dimensions are not mutually exclusive. Fatigue may be mental or physical while also occurring acutely or chronically over time [14,20]. Additionally, fatigue may arise during active or passive circumstances [54]. 

Within the Pharmacist Fatigue Instrument, 52% of pharmacists reported they had feelings where they could not keep up with their work most days. This may relate to the feelings of active fatigue expressed during the interviews where participants stated they were fatigued after helping “person after person”. Similarly, over 32% of pharmacists reported forgetting whether they had completed tasks on most of the days they worked. This may align with feelings of passive fatigue in the interviews or pharmacists that stated they felt they were working on “autopilot.”

The Adaptation Model of Stress and Performance, including active and passive fatigue, help to explain unique scenarios of fatigue, namely why under-stimulating work can be just as exhausting as over-stimulating work. A study by Saxby et. al. identified that when simulating active and passive fatigue in drivers (wind gusts inducing active fatigue and full vehicle automation inducing passive fatigue), only passive fatigue reduced alertness, speed of braking, and steering responses to an emergency event [56]. 

The pharmacists in this study highlighted the fine balance between staying busy enough to stay engaged in work while also being too busy to slow down or recover between patients. This idea of balance can be challenging in community pharmacy, and other retail professions, especially because they are unable to control their rate of work [55]. Patients do not make appointments to pick up their prescriptions, leaving some times where pharmacies are “slow” and other times where pharmacists are inundated with patients, prescriptions, and potential problems. “Solving” these problems of active and passive fatigue are challenging because they require changes to entire community pharmacy work systems. For example, pharmacies moving to an appointment-based model or medication synchronization programs may help to control when patients enter the pharmacy [57]. Task delegation or shifting to expand the role of certified pharmacy technicians, including tech-check-tech or technician-provided immunizations, may be helpful to ensure that pharmacists have the ability to focus on clinical tasks [58,59]. 

Ultimately, pharmacist occupational fatigue is a complex multidimensional phenomenon. Solutions to address occupational fatigue must consider how to address the mental, interpersonal, as well as work system components of pharmacist experiences.

### 4.4. Limitations and Future Directions

This study faces potential selection bias, in that individuals experiencing high levels of fatigue may be more interested in participating. Conversely, pharmacists who feel overwhelmed and unable to keep up with work may not have the time or energy to participate in a multi-week study. Recruitment mainly included pharmacists engaged in the PearlRx practice-based research network, which may limit generalizability of findings.

There are inherent limitations to the Pharmacist Fatigue Instrument in that it assigns a continuous value to a non-continuous/categorical response variable, potentially inflating the results. However, this approach has been used in other fatigue instruments, including the Swedish Occupational Fatigue Inventory and follows the assumption that normally distributed, continuous variables underly the categorical variables of the Likert-scale [14,60,61]. 

The Pharmacist Fatigue Instrument has been shown to demonstrate the mental and physical domains of occupational fatigue—it has not explicitly been tested to capture interpersonal relationships or components of active and passive fatigue. Therefore, researchers should be cautious in using this instrument alone to distinguish discrete fatigue domains. The Pharmacist Fatigue Instrument may be paired with other validated fatigue instruments such as the Swedish Occupational Fatigue Inventory or the Occupational Fatigue Exhaustion/Recovery Scale to isolate specific fatigue domains or compare fatigue in pharmacists or generalize to other professions. However, the Instrument presents a holistic survey to capture experiences of occupational fatigue tailored specifically to community pharmacists. 

Future research may be done to expand the use of the Pharmacist Fatigue Instrument to pharmacists in setting outside of community pharmacy as well as pharmacy technicians or student interns. Future studies may also explore the validity of the Pharmacist Fatigue Instrument at comparing fatigue before and after the implementation of interventions aimed at improving pharmacist fatigue.

## 5. Conclusions

The goal of this study was to describe Wisconsin community pharmacists’ experiences with occupational fatigue. This study is valuable in that it took a multimodal approach of a tailored Pharmacist Fatigue Instrument and semi-structured interviews. The Pharmacist Fatigue Instrument may be useful for researchers or organizations wishing to capture a holistic picture of community pharmacist fatigue. Interventions aimed at improving occupational fatigue in community pharmacies should consider key themes of fatigue that pharmacists are experiencing including mental fatigue, interpersonal connections and relationships, as well as the complex work system impact of active and passive fatigue.

## Figures and Tables

**Figure 1 pharmacy-11-00084-f001:**
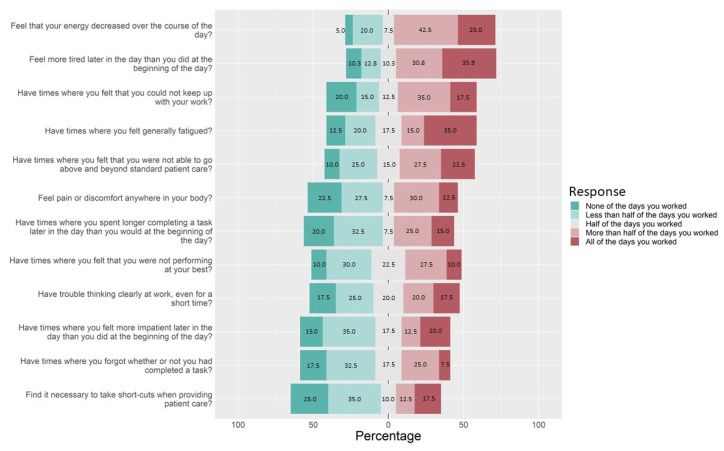
Pharmacist Fatigue Instrument.

**Figure 2 pharmacy-11-00084-f002:**
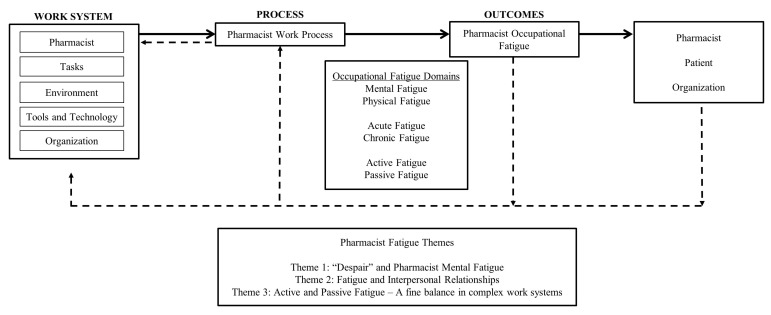
Pharmacist Occupational Fatigue Conceptual Model and Key Interview Themes.

**Table 1 pharmacy-11-00084-t001:** Participant Demographics.

Characteristics	N = 39 (%)
Pharmacy Characteristics	
Practice Setting	
National Chain	14 (35.0%)
Independent	15 (37.5%)
Other (Mass Merchandiser, Grocery Store, Other)	11 (27.5%)
Scheduled Breaks in the Workday	
Yes	28 (70.0%)
No	12 (30.0%)
Average Characteristics	Median (range)
Days Worked in the 14-day Period	10 days (6–14)
Shift Length	9 h (7–12.5)
Pharmacist Support	2 full-time pharmacists (1–10)
Staff Support (Technician, Student Intern, Clerk)	3.5 full-time staff members (0–10)
Prescription Volume	400 prescriptions (17–1500)
Pharmacist Characteristics	
Gender Identity	
Man	12 (30.8%)
Woman	26 (66.7%)
Prefer to Self-Describe	1 (2.6%)
Ethnicity	
White, not of Hispanic Origin	33 (84.6%)
Asian or Pacific Islander	7 (17.9%)
Marital Status	
Married	23 (59.0%)
Single	13 (33.3%)
Divorced	3 (7.7%)
Average Characteristics	Median (range)
Age	34 years old (25–59)
Daily Caffeine Servings	2 servings (0–4.5)
Adults (age 18+) in the Household	2 adults (1–5)
Individuals (under age 18) in the Household	0 individuals (0–3)
Years of Working in the Pharmacy as a Pharmacist	3 years (0–31)
Hours of Sleep	7 h (5–9)

## Data Availability

The data underlying this article cannot be shared publicly for the privacy of individuals that participated in the study. Data will be shared on reasonable request to the corresponding author with permission of the University of Wisconsin—Madison.

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
