# Peer review of "Subjective Perceptions of Occupational Fatigue in Community Pharmacists"

_pharmacy, 2023, doi:10.3390/pharmacy11030084_

Round 1

Reviewer 1 Report

Thank you for the opportunity to review this paper on community pharmacists' perceptions about fatigue, a very timely topic to the profession, and I think a valuable addition to the literature. I have a few minor comments for your consideration. 

Title: I would say that perceptions are inherently subjective and thus feel the inclusion of the word is redundant

Page 2, line 50-51 - I re-read several times because I didn't understand the list of domains. Can you provide additional context other than just the list?

Page 8, line 318 - what reference do you have to prove their is a pharmacist shortage? More likely, there is a shortage of pharmacist work hours available from major corporations contributing to a shortage of pharmacists willing to work under these conditions. Consider removing this statement or finding a reference to corroborate.

Line 339 - "algin" should be "align"

Line 345 - I believe "inducting" should be "inducing"?

Author Response

Thank you so very much for taking the time to review the manuscript.

Reviewer Comment

Response

Title: I would say that perceptions are inherently subjective and thus feel the inclusion of the word is redundant

Thank you for your comment! I’m unsure if there’s a way to easily change the title of the manuscript. At this time, we will keep the work “Subjective” but if the editorial team chooses to remove, we happily agree.

Page 2, line 50-51 - I re-read several times because I didn't understand the list of domains. Can you provide additional context other than just the list?

Thank you! Elaborated by provided descriptions of various types of fatigue.

Page 8, line 318 - what reference do you have to prove their is a pharmacist shortage? More likely, there is a shortage of pharmacist work hours available from major corporations contributing to a shortage of pharmacists willing to work under these conditions. Consider removing this statement or finding a reference to corroborate.

Thank you for your comment! The pharmacist shortage reference came from a New York Times article describing the pharmacy “labor shortage.”

You provide a comment that this is likely a shortage of pharmacists [and technicians] willing to work under strained working conditions.

Ultimately, we removed the statement.

Link: https://www.nytimes.com/2022/02/10/us/politics/pharmacists-strain-covid.html

Line 339 - "algin" should be "align"

Thank you.

Line 345 - I believe "inducting" should be "inducing"?

 Thank you.

Reviewer 2 Report

Community Pharmacist Subjective Perceptions of Occupational Fatigue

Overall, this is an article about an important concept.  We must look at pharmacist fatigue (and pharmacist burnout in general) to move the profession forward.  The overall recommendation I have is to clarify the results.  Right now, it seems that new information and framing is presented in the discussion.  The results are framed by the type of occupational fatigue, but then three themes are presented in the discussion.  Consider framing the results in the context of these three themes.

Introduction:

-       In both the abstract and the introduction, the authors note that “One aspect of workload stress that has been dangerously overlooked among pharmacists is occupational fatigue.”  The wording of “dangerously overlooked” seems a bit unclear to me.  Perhaps the authors could omit the word “dangerously” or further explain the details of the danger.

-       Line 51: Can the authors elaborate on the list of different types of occupational fatigue?  I think it would be nice for the readers to have an understanding of what these terms mean in the context of occupational fatigue.

Recruitment:

-       Please include more information about PearlRx.  Could the pharmacists in this program have different backgrounds and/or perceptions of fatigue compared to pharmacists not in the program?

Results:

-       Previous work has indicated that pharmacist burn out rates differ by work location. Did this study look at correlations between fatigue rates and work setting (for instance, chain vs. independent)?

Discussion:

-       The authors bring up three themes from the instrument and the interviews.  These are likely best presented in the results. It might be easy for the reader if the authors are able to provide a theoretical framework, perhaps with a figure, in the results that demonstrates their three themes.

-       The authors highlight that over 50% of participants were more inpatient later in the day.  What is the expected finding for other professionals? It seems that this finding might be expected?

-       Similar to the comment above, what is the result around going above and beyond for other professions? 

-       The authors bring up teamwork in the discussion around systems issues related to workload and chronic fatigue.  They might consider referencing Shanafelt’s work out of Mayo or the National Association of Medicine’s work around teamwork and burnout.  I’ll list the references here:

o   Shanafelt TD, Noseworthy JH. Executive Leadership and Physician Well-being: Nine Organizational Strategies to Promote Engagement and Reduce Burnout. Mayo Clin Proc. 2017 Jan;92(1):129-146. doi: 10.1016/j.mayocp.2016.10.004. Epub 2016 Nov 18. PMID: 27871627.

o   Smith, C. D., C. Balatbat, S. Corbridge, A. L. Dopp, J. Fried, R. Harter, S. Landefeld, C. Martin, F. Opelka, L. Sandy, L. Sato, and C. Sinsky. 2018. Implementing optimal team-based care to reduce clinician burnout. NAM Perspectives. Discussion Paper, National Academy of Medicine, Washington, DC. https://doi.org/10.31478/201809c

-       In the discussion, the authors state that “This research study demonstrated that fatigue is related to pharmacist interpersonal relationships with family, team members, and patients.” Can this be brought out in the results more? In particular, I don’t see much mention of relationships with family members prior to this statement.  It would be best to clarify this in the results.

Author Response

Thank you so much for taking the time to review our manuscript and for the fantastic comments. You truly helped to make this manuscript better! 

Reviewer Comment

Response

Introduction

In both the abstract and the introduction, the authors note that “One aspect of workload stress that has been dangerously overlooked among pharmacists is occupational fatigue.”  The wording of “dangerously overlooked” seems a bit unclear to me.  Perhaps the authors could omit the word “dangerously” or further explain the details of the danger.

Thank you for your comment. We omitted the word “dangerously.” Our intention was to highlight the safety concerns when pharmacists are fatigued (e.g., increased risk for error, needle-stick injuries, car accidents on the ride home, etc.), but you are correct, it is somewhat subjective.

Line 51: Can the authors elaborate on the list of different types of occupational fatigue?  I think it would be nice for the readers to have an understanding of what these terms mean in the context of occupational fatigue.

Thank you! We elaborated and provided descriptions of various types of fatigue.

Recruitment

Please include more information about PearlRx.  Could the pharmacists in this program have different backgrounds and/or perceptions of fatigue compared to pharmacists not in the program?

Thank you! We added more detail regarding PearlRx to the recruitment section. While PearlRx represents a diverse range of pharmacist work settings, we added an additional statement to our limitation section—pharmacists who are able and willing to participate in PearlRx may have differing levels of fatigue than those who do not participate (e.g., those who are able to participate in research/PearlRx may experience less fatigue in general).

Results

Previous work has indicated that pharmacist burn out rates differ by work location. Did this study look at correlations between fatigue rates and work setting (for instance, chain vs. independent)?

Thanks for your comment! We were also interested in whether fatigue may differ across different work settings: Chain, Independent, and Other!

Regarding the Pharmacist Fatigue Instrument:

Physical and mental fatigue items were aggregated, and each participant received a Physical Fatigue Total and Mental Fatigue Total. For the physical fatigue domain, the total possible score for the 3 items was 12. For mental fatigue, the total possible score for the 4 items was 16. A higher total Physical Fatigue score indicated that Physical Fatigue events occur “more frequently” than a lower Physical Fatigue score. 

Total Physical and Mental Fatigue scores were compared across pharmacy practice settings using a one-way Analysis of Variance (ANOVA) statistical test. The ANOVA tested the hypotheses that there was no statistical difference in Physical and Mental Fatigue totals based on pharmacy practice setting.

The results were not only non-significant, but we found that the results didn’t really “mean” much when trying to compare quantitatively.

The qualitative findings did not differ significantly across the different pharmacy settings.

Discussion

The authors bring up three themes from the instrument and the interviews.  These are likely best presented in the results. It might be easy for the reader if the authors are able to provide a theoretical framework, perhaps with a figure, in the results that demonstrates their three themes.

Thank you for your comment. We presented our framework in the results section for how we conceptualize occupational fatigue in pharmacists (we also added a brief sentence to the methods to state the background literature).

Currently, our entire Occupational Fatigue Conceptual Model is under review in a separate publication, so it’s important to distinguish that the results from this publication only pertain to the “Outcome” of pharmacist fatigue.

We found it important to keep the results from the themes. The results were related explicitly to the occupational fatigue domains. In the discussion section, we wished to extrapolate further to the literature, as well as combine some of the quantitative and qualitative findings into more generalizable themes.

We hope that the inclusion of the conceptual model and figure helps to orient the reader to the results and the discussion sections.

The authors highlight that over 50% of participants were more inpatient later in the day.  What is the expected finding for other professionals? It seems that this finding might be expected?

Thank you for your comment. We agree, the finding is somewhat expected. The desire to ask about ‘feeling impatient’ came about during original instrument development. Participants reported they knew they were fatigued when they were more irritable and short towards their staff and patients.

Although a direct comparison of “impatience” in medical professionals isn’t found in the literature, a study of emergency medicine consultants found that approximately one-third reported work-stress irritability. We’ve added this study to the discussion,

The sentiment behind the statistic is to further demonstrate that fatigue is larger than an individual phenomenon.

Similar to the comment above, what is the result around going above and beyond for other professions? 

Thanks for the great question! Similar to the above response, there’s no direct correlation to studies found in the literature regarding healthcare professionals “going above and beyond” when fatigued (we added a citation to the above mentioned study in which emergency medical consultants reports a reduced standard of care).

The question came about during instrument development/developmental interviews when participants would describe how they didn’t feel “unsafe” when fatigued but they didn’t go out of their way to help patients. For example, instead of leaving the pharmacy counter to help a patient find a product, they would point or direct the patient verbally.

Pharmacists told us that they felt that it wasn’t providing “bad” care, they simply provided “better” care when they weren’t fatigued.

The authors bring up teamwork in the discussion around systems issues related to workload and chronic fatigue.

Thank you so very much for these great references! We’ve added them to the discussion.

In the discussion, the authors state that “This research study demonstrated that fatigue is related to pharmacist interpersonal relationships with family, team members, and patients.” Can this be brought out in the results more? In particular, I don’t see much mention of relationships with family members prior to this statement.  It would be best to clarify this in the results.

Thank you! We added clarification to the results, explicitly calling out the impact of mental fatigue on pharmacists relationships at home.

Reviewer 3 Report

This is a well-conducted and accurately reported research on pharmacists' perceptions on occupational fatigue. For this purpose two different tools were used: the Pharmacist Fatigue Instrument and semi-structured interviews. The Introduction is comprehensive. Results are clearly presented and study limitations as well as study strength are summarised. 

I have few questions for the authors: 

The study aimed to recruit 60 community pharmacists, however just 39 participants were part of data collection. How was the sample calculated? Why didn´t the authors consider other ways for recruitment? Is this number of participants in the study appropriate to support conclusions and generalise results? 

Author Response

Thank you so very much for taking the time to review our manuscript and for your thoughtful questions.

Reviewer Comment

Response

The study aimed to recruit 60 community pharmacists, however just 39 participants were part of data collection. How was the sample calculated? Why didn´t the authors consider other ways for recruitment? Is this number of participants in the study appropriate to support conclusions and generalise results?

Thank you! We added more detail regarding PearlRx to the recruitment section as well as information regarding our sample size selection.

Our team did use other methods of convenience sampling to encourage recruitment. Individuals who participated asked if they could forward the study to other pharmacists or colleagues in their workplace, which we happily encouraged. The project was also posted publicly on the PearlRx “Current Projects” page.

Recruitment dwindled as we reached n=39. Our qualitative findings reached saturation and our team felt confident that our results were representative of Wisconsin community pharmacists. We were also limited by financial and time restrictions for the project.

We added a statement regarding our recruitment limitations to the discussion section.